# X-ray Phase Contrast Imaging from Synchrotron to Conventional Sources: A Review of the Existing Techniques for Biological Applications

**Laurene Quenot** [ID], **Sylvain Bohic** [ID] and **Emmanuel Brun** *[ID]

INSERM, UA7 STROBE, Synchrotron Radiation for BioMedecine, Grenoble Alpes University, 38000 Grenoble, France
* Correspondence: emmanuel.brun@inserm.fr

**Featured Application: X-ray phase contrast and dark-field imaging have been recently introduced in conventional systems. They have been demonstrated to outperform the conventional imaging modalities. This article proposes a comprehensive overview of the experimental techniques that allow obtaining these new contrasts on conventional systems.**

**Abstract:** Since the seminal work of Roentgen, X-ray imaging mainly uses the same physical phenomenon: the absorption of light by matter. Thanks to third-generation synchrotrons that provide a high flux of quasi-coherent X-rays, we have seen in recent years new imaging concepts such as phase contrast or dark-field imaging that were later adapted to conventional X-ray sources. These innovative imaging techniques are particularly suitable for visualizing soft matter, such as biological tissues. After a brief introduction to the physical foundations of these two techniques, we present the different experimental set-ups that are now available to produce such contrasts: propagation, analyzer-based, grating interferometry and non-interferometric methods, such as coded aperture and modulation techniques. We present a comprehensive review of their principles; associated data processing; and finally, their requirements for their transfer outside of synchrotrons. In conclusion, gratings interferometry, coded aperture and modulation techniques seem to be the best candidates for the widespread use of phase contrast and dark-field imaging on low-cost X-ray sources.

**Keywords:** X-ray imaging; phase contrast; darkfield; gratings interferometry; edge illumination; speckle based imaging; modulation based imaging

## 1. Introduction

During the last three decades, X-ray phase contrast and dark-field imaging (PC and DI) have been proposed to overcome the limitations of absorption-based imaging in two directions: increasing contrast for soft tissue and reducing the radiation dose. Since the seminal work of Roentgen, X-ray imaging was based on the absorption phenomenon, but refraction, the fact that light is deviated when passing through matter, shows promise. Indeed, the refractive index of the material can be a thousand times greater than its absorption counterpart for light elements. This translates into a much greater contrast for all the different tissues with X-ray imaging methods based on phase sensing. A third type of contrast, named dark-field, is sensitive to multiple refractions and gives access to sub-pixel information, which is very valuable for lung imaging, for instance.

Over the past few decades, an increasing number of studies have demonstrated the high diagnostic potential of PC and DI [1], as compared to conventional radiology, in a wide range of pathologies and applications, including mammography [2,3], osteoarticular diseases [4,5], brain [6–8] and pulmonary diseases [9,10]. With the emergence of partially coherent X-ray sources twenty years ago, expectations regarding PC and DI became feasible,

following which several PC and DI methods have been developed at synchrotrons. Unfortunately, due to the dimensions and cost of such infrastructures, synchrotrons cannot be used for the clinical routine for widespread pathologies. Therefore, several teams worldwide are working on the transfer of X-ray PC and DI to conventional systems more adapted to the clinical constraints.

Multiple techniques and experimental set-ups have been developed over the past 25 years to exploit phase-contrast in the X-ray regime. The goal of this review article is to present the most promising experimental set-ups for the transfer of these techniques. After introducing the physical foundations of these new contrast, we present in this article the five main techniques for producing X-ray PCDI images on conventional systems with an emphasis on their computed tomography capabilities.

## 2. Physical Foundations of X-ray Phase Contrast and Dark-Field Imaging

Phase contrast and dark-field imaging consider X-ray light interactions with matter from a wave perspective. For extensive comprehension of the physical phenomenon implied in X-ray phase contrast imaging, we refer the reader to [11,12].

An X-ray passing through matter interacts with the electronic clouds and nuclei of atoms, more or less strongly depending on the atomic numbers of the atoms. Interactions are described by the refractive index: $n = 1 - i\beta - \delta$. $\delta$ accounts for phase shift (i.e., refraction) and $\beta$ for attenuation. $\beta$ is directly related to the linear attenuation coefficient $\mu = 2k\beta$. $\delta$ and $\beta$ vary with the sample electronic density and the incident photons energy E (in $1/E$ for $\beta$ in the X-ray range of energy and $1/E^3$ for $\delta$).

### 2.1. Theoretical Coherent Case

To introduce the first equations and phenomena, let us assume we are in a monochromatic case where the wave $\psi_\lambda$ is emanating from a point-like source at an infinite distance and that the beam is parallel to the optical axis. As described in [11], this wave evolves when propagating through free space following the free space paraxial equation:

$$(2ik\frac{\partial}{\partial z} - \nabla_\perp^2)\psi_\lambda(x,y,z) = 0 \tag{1}$$

$k$ is the wavenumber and $\lambda$ is the wavelength ($\lambda = 2\pi/k$). Diffraction of the wave over distance $z_1$ can then be written as:

$$\psi_\lambda(x,y,z_0+z_1) = D_{z_1}^{(F)}\,\psi_\lambda(x,y,z_0) \tag{2}$$

where $D_{z_1}^{(F)}$ is called the Fresnel propagator and is written as an operator:

$$D_{z_1}^{(F)} = \exp(ikz_1)\mathcal{F}^{-1}\exp\left(-\frac{iz_1}{2k}(k_x^2+k_y^2)\right)\mathcal{F} \tag{3}$$

That was for free space propagation. Now, what happens when the wavefield encounters an object? In the case of thin samples that are sufficiently slowly varying in space, we can use the projection approximation, which allows us to consider that the X-ray beam follows a straight line within the sample. The projection approximation validity is discussed in [12,13]. Under the approximation and using the complex refractive index introduced earlier, we describe two quantities: the phase shift ($\Delta\phi$) and attenuation ($B$) through the sample:

$$\Delta\phi_\lambda(x,y) = -k\int_z \delta_\lambda(x,y,z)dz \qquad B_\lambda(x,y) = k\int_z \beta_\lambda(x,y,z)dz \tag{4}$$

They are then combined as follows to compute the wave distortions through the sample:

$$\psi_\lambda(x,y) = \psi_{0,\lambda}(x,y).\exp(i\Delta\phi_\lambda(x,y) - B_\lambda(x,y)). \tag{5}$$

where $\psi_{0,\lambda}(x,y)$ is the scalar wave-field before the object. An important quantity that should be introduced at that point is the intensity of the wave-field: $I_\lambda(x,y,z) = |\psi_\lambda(x,y,z)|^2$. This quantity is the one that will be ultimately measured by the detector in X-ray imaging. The relation between the intensity and the wave-field allows us to retrieve the well known Beer–Lambert Law, which gives the attenuation through an object:

$$
\begin{aligned}
I_\lambda(x,y,z_{object}) &= |\psi_\lambda(x,y,z_{object})|^2 \\
&= I_\lambda(x,y,z_0)\exp(-2B_\lambda(x,y)) \\
&= I_\lambda(x,y,z_0)\exp\left(-\int_z \mu_\lambda(x,y,z)dz\right)
\end{aligned}
\tag{6}
$$

Under the assumption of local conservation of the optical energy, one can describe the evolution of the intensity upon propagation in free space with the transport of intensity equation (TIE) [14]:

$$
-\nabla_\perp.[I_\lambda(x,y,z)\nabla_\perp\phi_\lambda(x,y,z)] = k\frac{\partial I_\lambda(x,y,z)}{\partial z}
\tag{7}
$$

This equation is fundamental in many phase contrast imaging methods.

Now let us see what happens in more complex cases, when the beam is no longer parallel and is less coherent. We focus on the simple imaging case where we have a source, an optical element and a detector. It can easily be extended to several optical elements, as is the case in many phase contrast imaging set-ups. In this simple case, the wavefield propagates in free space from the source to the object over a distance $z_1$, then gets distorted in the object and propagates again in free space to the detector over a distance $z_2$. In the case of a divergent beam (or a convergent one, but we focus on the first case here), the wave distortion over distance can be calculated using the Fresnel scaling theorem. In this divergent case, we have a spherical wave and a magnification phenomenon. The magnification is simply calculated using: $M = (z_1 + z_2)/z_2$. The Fresnel scaling theorem states that the Fresnel diffraction patterns due to a sample in a cone beam, observed after a distance $z_2$, are equivalent to the one that would be observed at a distance $z_2' = z_2/M$ in a parallel beam.

### 2.2. Partially Coherent Systems

To date, we have seen the theoretical case of a coherent monochormatic point-like source. In reality, monochromatic sources are limited to large facilities such as synchrotrons. These also provide small emitting spots and can be considered quasi-coherent. However, more conventional easy-access sources have less advantageous features. Firstly, they are polychromatic; i.e., they emit over a large bandwidth of energies. Secondly, their emitting spots are quite spread out (from 50 to 200 μm), meaning that they lack spatial coherence. What are the consequences for the image formation? First, the wave-front varies differently for each wavelength, and the intensity of each wavelength is weighted by the source emission spectrum. In the end, the total intensity can be represented as the integral of the propagated wavefront over the source spectrum:

$$
I(x,y) = \int_\lambda I_\lambda(x,y)w_\lambda d\lambda = \int_\lambda |\psi_\lambda(x,y)|^2 w_\lambda d\lambda
\tag{8}
$$

with $w_\lambda$ being the weight of the wavelength in the source emission spectrum. Then, the source focal spot will induce a blurring of the image. As the photons are not all emitted from one single point, photons hitting one point of the sample will indeed have slightly different incident trajectories, and therefore, not hit the exact same spot on the detector, thereby introducing blurring of the image. This phenomenon can be seen as a convolution of the resulting image with the projected shape of the source on the detector plane, taking into account the distances $z_1$ and $z_2$. These two phenomena result in a blurring of the image

that forms due to wave alterations in the sample, making different signal extractions more complicated with such poor coherent sources.

What are the signals that we are talking about? From those intensity and wave alterations in the sample, three main phenomena can be observed when imaging a sample with X-rays: attenuation, phase shift and small angle scattering (see Figure 1).

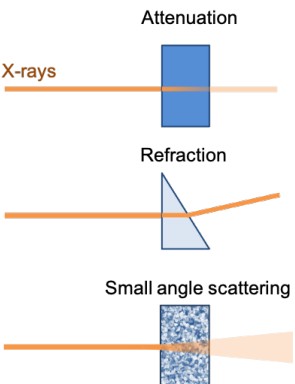

**Figure 1.** Three main phenomena observed when imaging a sample with X rays.

### 2.3. Attenuation

The easiest to observe is the attenuation mainly due to the photoelectric effect and Compton scattering. It appears directly in the image as a difference in intensity and is described by the Beer–Lambert Law (Equation (6)). This is the phenomenon used in conventional radiography and clinical CT.

### 2.4. Phase Contrast Imaging (PCI)

The second one is phase shift (or refraction) of resolved structures. This phenomenon can appear as interference fringes in systems that are coherent enough but is invisible in conventional radiography images. From the Snell–Descartes law, a simple relation between the refraction angle and the phase shift can be established:

$$\alpha(x,y) = \frac{1}{k}\nabla_\perp \phi(x,y) \tag{9}$$

The refraction angles of X-rays are in the micro-radians range, making them very difficult to observe. However, the refractive index of light element materials can be a thousand times greater than its counterpart, the absorption factor, for the wavelengths used in radiology [1]. This induces a much greater contrast for soft tissues with X-ray imaging methods based on the detection of the light refraction or with a wave description of the phase [15]. Different groups around the world used this principle since the mid of 1990s to increase the contrast for soft tissues in various cases [16,17] or used the very same principle to decrease the radiation dose while giving reasonable image quality and even improved quality compared to the conventional absorption-based radiography [3,18].

### 2.5. Dark-Field Imaging (DI)

The last phenomenon that can be observed is small angle X-ray scattering (SAXS) or ultra-small angle scattering (USAXS). It is also due to refraction but of numerous unresolved structures. The multiple refraction, happening in a medium composed of dense sub-pixel sized micro-structures invisible directly on the image, induces fan spreading of the beam. A study from 1926 [19] gives an estimate of the average deflection angle of the rays from the number of particles crossed (Npart) and the difference of refractive index between the environment and the particles:

$$\alpha(x,y) = \sqrt{4\delta^2 N_{part}(x,y)(log\frac{2}{\delta} + 1)} \tag{10}$$

This phenomenon appears on images as local blurring, and numerical processing is required to access that information. X-ray dark-field imaging was only very recently developed, but its interest for the study of lung diseases was rapidly demonstrated because it is an indicator of the alveoli's state of health [20].

In order to retrieve these different signals, conventional imaging set-ups have to be altered. This review article presents different phase contrast imaging techniques, their main principles, their requirements and their progress toward clinical implementation. We will try to describe them with homogeneous terms and equations for better understanding. Let us call $I_R(x, y)$ the reference image collected in the absence of a sample. This image, the absence of an optical element is the image of the beam, also known as the white field. Let us call $I_S(x, y)$ the sample image collected in the same geometrical conditions but with a sample. This image is therefore a projection of the sample with or without an optical element depending on the set-up. Let us call $I_r^i(x, y)$ $I_s^i(x, y)$ the couple of reference and sample images taken at the $i^{th}$ position of an additional optical element. In this manuscript, we call $\lambda$ the wavelength and $\delta$ the real part of the refractive index: $\delta = \frac{Nr_c\lambda^2}{2\pi}$ with $N$ the electron density and $r_c$ the absorption cross section. We also call $\beta$ the imaginary part of the refractive index representing the absorption. $\beta = \frac{\lambda\mu}{4\pi}$ where $\mu$ is the linear mass attenuation coefficient. $\gamma$ will be the ratio $\frac{\delta}{\beta}$. Finally, we call $\phi$ the phase, then $\Delta\phi = \frac{2\pi\delta T}{\lambda}$, where $T$ is the thickness of a material along the $z$ propagation axis.

## 3. Phase Contrast Imaging Techniques on Conventional Systems

While attenuation is directly visible on simple acquisitions, the phase and dark-field appear more as small artifacts in the images, and retrieving their signal is not straightforward. PCI and DI techniques differ by their experimental set-ups but also by their associated data processing methods. They require different spatial and temporal coherence levels, and due to their optical set-ups, they are more or less optimized for radiation dose deposition. The following paragraphs present the physical principles, the contrast mechanisms, the associated phase retrieval methods and finally, the advantages of all these techniques. In the following sections, we will describe only the latter ones that can be divided into five categories:

1. Propagation-based imaging (PBI) [21]
2. Analyzer-based imaging (ABI) [15].
3. Grating interferometry (GI) [22].
4. Edge illumination (EI) [23].
5. Mesh-based imaging (MBI) [24].
6. Modulation-based imaging techniques (MoBI) [24–26].

### 3.1. Propagation-Based Imaging

Propagation-based imaging (PBI) is the most used experimental set-up in synchrotrons around the world. Its success is no doubt due to its very simple experimental set-up, the rapidity of the acquisition and the simple numerical phase retrieval processing.

As presented in Figure 2, the key features of this method are the coherence of the source and the propagation distance between the sample and the detector. Apart from that, the set-up is similar to one of conventional radiography.

The only difference with a standard clinical X-ray imaging system is the fact that the detector is separated from the sample by a distance that depends on the desired spatial resolution. The principle is that, thanks to the spatial coherence of the source, upon propagation, the waves distorted in the sample will create interference patterns that are more and more prevalent as the distance between the sample and the detector increases. Those interference fringes appear as white and black lines contouring the sample; it is what we call "edge enhancement". This interference pattern, appearing as intensity variations in the acquisitions, is proportional to the phase Laplacian of the waves distorted by the sample.

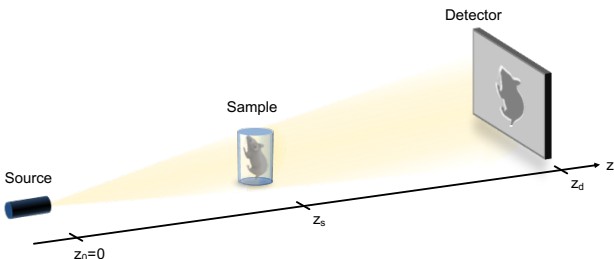

**Figure 2.** Propagation-based imaging set-up.

To retrieve the phase information, various methods have been developed and optimized around the world, but the fundamental one is the so-called "Paganin method" [27]. It is based on the transport of intensity equation (TIE) in the paraxial approximation (see Equation (11)).

$$-k\frac{\partial I(x,y)}{\partial z} = \nabla_\perp \cdot [I(x,y)\nabla\phi(x,y)] \tag{11}$$

$x$ and $y$ are the spatial coordinates in the plane perpendicular to the propagation axis $z$. An extensive development of this equation, including the derivation of the equation and its use with different approaches, can be found in the very well written review by Zuo et al. [28]. The resolution of this equation to extract the phase information proposed by Paganin et al. in [27] assumes a single material sample. This approximation allows one to assume constant $\delta$ and $\beta$ to solve the equation from only one acquisition. This assumption implies that the phase is proportional to the thickness $t$: $\phi = \delta t$. The equation to extract the thickness from the TIE then becomes:

$$t(x,y) = -\frac{1}{\mu}log\left(\mathcal{F}^{-1}\left\{\frac{\mathcal{F}\{I_s(x,y)/I_r(x,y)\}}{1-(z_2\delta/\mu)(k_x^2+k_y^2)}\right\}\right) \tag{12}$$

where $I_s$ is the sample image acquired with the sample, and $I_r$ is the reference image corresponding here to the white field without the sample, used for normalization.

Due to the strong single material hypothesis, the retrieved image does not give a quantitative measurement; however, it still gives a very good contrast between various materials.

In order to get free from this assumption and get quantitative results, a variant of this method was invented with several propagation distances $z$ (between the sample and the detector as shown in Figure 3) and based on the contrast transfer function (CTF) [29]:

$$\phi(x,y) = \frac{1}{I_r}\mathcal{F}^{-1}\left\{\frac{\sum_z\left[I_s^{(z)} - I_r - G^{(z)}(\mathbf{f})\mathcal{F}\{\nabla_\perp(\phi(x,y)\nabla_\perp I_r)\}\right]H^{(z)}(\mathbf{f})}{\sum_z\left(H^{(z)}(\mathbf{f})\right)^2}\right\} \tag{13}$$

where $\mathbf{f} = (f_x, f_y)$ is the Fourier space vector. Note that the variables used in Fourier space are sometimes the frequency noted $\mathbf{f}$ or $\mathbf{u}$ or the angular frequency $\mathbf{k} = 2\pi\mathbf{f}$. As much as possible, we try to harmonize the notation. It was chosen to keep the original article notation when possible: $G^{(z)}(f) = \frac{\lambda z}{2\pi}cos(\pi\lambda z f^2)$ and $H^{(z)}(f) = 2sin(\pi\lambda z f^2)$. In this method, the equation is solved iteratively, neglecting the term containing the phase for initialization.

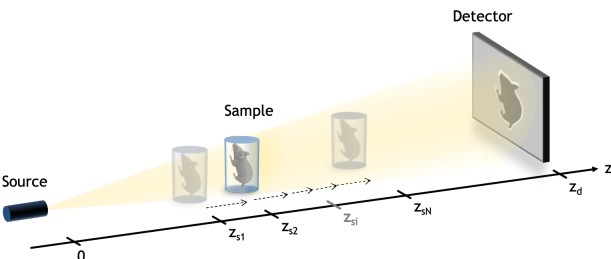

**Figure 3.** Holography imaging set-up. A coherent beam traverses the sample and propagates a certain distance before reaching the detector. Interference fringes due to wave perturbation in the sample appear on the detector plane and vary according to the distance of propagation. Several images with various distances of propagation or only one position can be taken.

Due to the high coherence needed, the transfer of this technique to other sources than a synchrotron was mainly limited to high-resolution imaging [7] using mostly liquid metal jet X-ray sources [30]. Imaging of patients can possibly be done with a high-energy compact source such as thomX [31] or MuCLS [32] projects. Nevertheless, the cost of such sources, and the fact that patients would have to be rotated for 3D imaging, will limit the PBI spread. Moreover, PBI is sensitive to the phase Laplacian that introduces reconstruction artifacts when imaging materials presenting slowly varying density and low spatial frequencies [1]. Finally, retrieving the dark-field from PBI images has just started to be investigated and appears to be limited to observing its effect at the edges of the sample [33,34].

### 3.2. Analyzer-Based Imaging

The first phase gradient sensitive technique, analyzer-based imaging (ABI), is also known as diffraction enhanced imaging, and it is based on filtering the transmitted beam through highly selective analyzer crystals.

Its experimental configuration consists of a monochromator upstream of the sample and an analyzer crystal positioned according to the Bragg geometry between the sample and the detector, as shown in Figure 4.

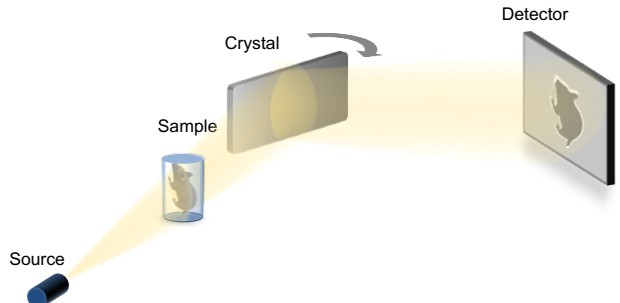

**Figure 4.** Analyzer-based imaging set-up. A monochromatic beam traverses the sample. Then it arrives on a Laue crystal that only reflects the beam arriving with a precise incident angle toward the detector. The crystal can be rotated to acquire several images with various reflection angles to retrieve the refraction from the sample.

This analyzer crystal acts as an angular filter for the radiation coming from the sample. When the X-rays are refracted by an object, the angle of incidence on the analyzer crystal is changed. When the X-rays reach the analyzer crystal, the Bragg diffraction condition is satisfied only for a small range of angles. Thus, when the scattered or refracted X-rays have incident angles outside this range, they are not reflected and do not contribute to the signal. By adjusting the tilt angle of the analyzer crystal, the refraction angle can be extracted.

This phase contrast experimental method proved to be the most sensitive [35] and feasible technique with low dose deposition to the sample, as demonstrated in [3].

The method has been successfully implemented with laboratory sources at the cost of extremely long exposure times due to the low flux after monochromatization [36]. Moreover, the beam stability is difficult to maintain, even at synchrotrons. These characteristics do not make this technique a good candidate for transfer to conventional sources, and very few works have been done in this direction.

### 3.3. Grating Interferometry

Grating-based imaging (GI) is an X-ray PCI technique based on the use of grating interferometers. Such gratings are usually made of Au or Si and typically measure around 5 cm; they have periods of a few microns. It has been successfully adapted from visible-light grating interferometry [37]. Fairly recent, this technique has become popular in many fields because of its good adaptability to conventional X-ray tubes. The principle of GI is a phenomenon called the Talbot effect (observed in 1836 by the English inventor Henry Fox Talbot), where a periodic wave repeats its pattern at a certain distances in the Fresnel regime.

Figure 5 presents a GI set-up on a conventional source. An X-ray grating interferometer consists of two gratings (G1 and G2); a detector; and in the case of conventional X-ray source, an additional grating G0 that acts as a collimator creating multiple small X-ray sources. G1 is usually a phase grating, and G2 is an absorption grating.

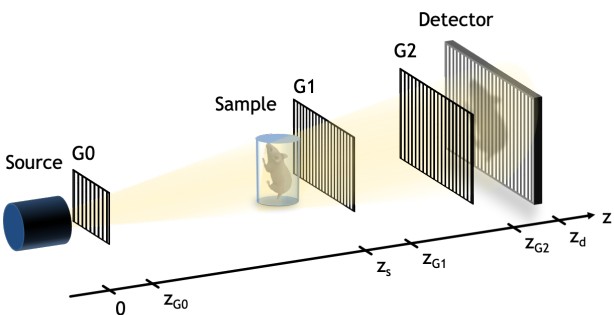

**Figure 5.** Grating interferometer imaging set-up. A beam from a very coherent source or from a source collimated by a G0 grating is emitted towards the sample. A grating G1 placed before or after the sample creates a Talbot carpet interference pattern. The grating G2 filters the peaks of intensity that have been deviated from the axis before the detector. Several acquisitions are required with various positions of G1 and G2.

G1 is used to split the beam, creating periodic interference patterns varying with distance. This interference pattern is called the "Talbot carpet", as shown in Figure 6. The interference pattern created from the waves re-emitted through G1 gives periodic peaks of intensity at characteristic distances. The detector is placed at one of these characteristic distances, usually where the intensity peaks are higher. When the sample is placed in the beam, these interference patterns will be distorted due to attenuation, refraction and scattering. When placing an attenuation grating in front of the detector with the same period as the original interference pattern, the phase variations are translated into intensity variations, as the peaks of intensity no longer fall in the space between two bars of G2. Taking several acquisitions with several positions of the gratings allows one to retrieve the precise refraction angles and the attenuation and spreading of the dark field.

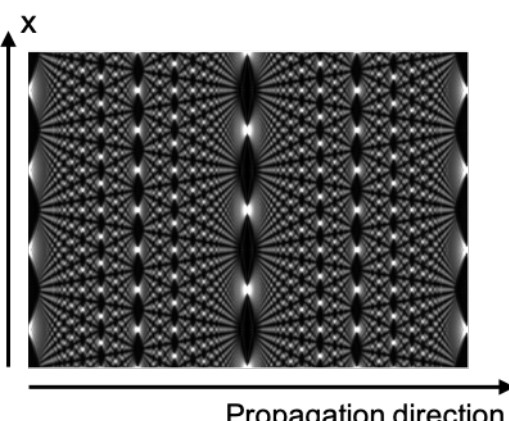

**Figure 6.** Example of a Talbot carpet pattern created by phase gratings.

There are two types of interferometers: the most common is a 1D interferometer that consists of gratings made up of parallel lines and the 2D grating interferometer designed with two-dimensional patterns [38]. Besides this latter proof of feasibility, it is difficult to find real application of these 2D interferometers. The fabrication of a gratings interferometer is a very challenging task and is a very active field of research [39–41], which is not the purpose of this review.

In 2006, Pfeiffer et al. [42] demonstrated the possibility of using the method on conventional sources, and a new field was born with tens of teams around the world working on this set-up. Albeit presenting a good sensitivity, this technique requires complex experimental set-ups difficult and is expensive to manufacture, especially for large fields of view. However, recent advances have shown impressive results on human patients using grating interferometry dark-field imaging [9,43,44].

Even more recently, the same team published the first results on the implementation of the technique on a clinical CT scanner [45]. This proof of concept on phantoms still needs to be optimized in order to reduce the acquisition time for clinical routine. Despite the set-up's precision requirement and stability, it was proven that, given the means and materials, it was possible to implement it on a tomography set-up.

The remaining obstacles that can be imputed to GI for complete 3D clinical adaptation are:

- Its limited dose efficiency as several positions of the gratings are required to obtain each projection, and part of the flux going through the sample will not be used to produce the image, as it will be absorbed in G2;
- Its acquisition time which is still too long for tomography;
- Its fabrication which is tedious and expensive;
- Its ability to retrieve the refraction only in one direction, making it insensitive to variations that are parallel to the gratings and causes bad performance in terms of noise [46] (noise power spectrum diverging with low frequency) with tomographic reconstruction due to a bad integration for the phase.

### 3.4. Edge Illumination

Initially developed as "coded aperture" [47] at synchrotrons and adapted to conventional sources [23], it has been called edge illumination (EI) since 2013 [48]. It is based on the observation that by illuminating only the edge of the detector pixels, high sensitivity to phase effects is obtained. The effect obtained is comparable to the ABI with a fine angular selection on the direction of the photons.

Figure 7 shows a typical edge Illumination set-up. The method uses a pair of masks, one before the sample and the other close to the detector. Those masks are usually made of Au or W with μm size wide apertures (slits) and tens of μm periods. Although the

configuration may appear similar to that of a grating interferometer, the physical principle is different. The spacing between the bars of the grids is wider than for GI, and instead of creating interference patterns, the grid placed in front of the sample (G0) simply splits the beam into thin vertical beamlets. In the absence of the sample, each of those beamlets will hit the center of each column of the detector. A second grid is placed in front of the detector, collimating vertically part of the pixels, reducing their individual fields of view. When the sample is inserted, the rays that are deviated will no longer arrive in the free spaces of G1, changing the intensity received by the detector. Displacing G1 slightly on one side and the other will allow us to retrieve the local refraction of the beam in the direction perpendicular to the grids and the dark-field scattering. The use of such a simple configuration, as opposed to ABI, eliminates the need for a monochromatic beam.

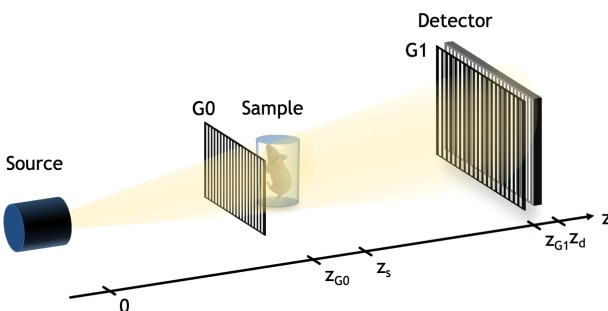

**Figure 7.** Edge illumination imaging set-up. A beam emitted toward the sample is first split into a thin line beamlet by a vertical grid G0. Those beamlets go through the sample before propagating to the detector plane, where a second grid G1 filters the beamlets that have been deflected by the sample. Several acquisitions are acquired with various positions of G0 and G1.

It has been shown that this technique requires only low spatial and temporal coherence. The method has been successfully implemented on conventional X-ray sources [49].

Though its set-up is simpler and the requirements in terms of coherence are smaller, EI suffers from some of the same limitations as GI for 3D: dose, acquisition time and difficulty to rotate around the sample/patient. This technique seems better suited for pre-clinical studies than in vivo radiological applications because of the long acquisition time/high dose delivered to the sample, since part of the radiation passing through the sample is stopped by the second slit. Another limitation remains the detection of the unilateral phase gradient only (even if a 2D implementation exists [50], their use remains scarce), and thus has the complexity to perform 3D imaging with this device.

### 3.5. Mesh-Based Imaging

In 2008, in the search of a technique requiring less exposure than GI and a simpler acquisition set-up, Wen et al. [24] invented spatial harmonic imaging. It has rapidly gone from a 1D to a 2D differential phase contrast modality [51]. The experimental set-up shown in Figure 8 consists simply of a 2D grid (or mesh) placed close to the sample in the path of the beam. The used meshes are adapted to the resolution of the system in order to have a period of typically 3 to 4 pixels and are usually made of woven steel. Only two acquisitions are needed, one with only the grid in the path of the beam to collect the reference image ($I_r$), and one with the sample added in the path of the beam ($I_s$). Ideally, the grid should entirely absorb the rays and be as thin as possible.

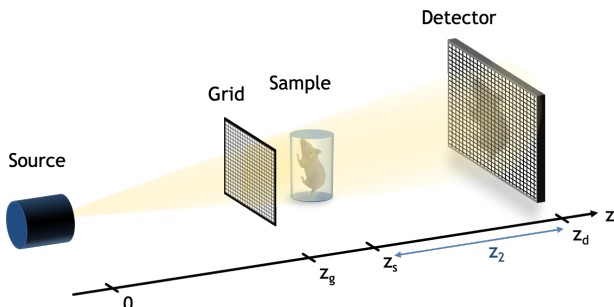

**Figure 8.** Mesh-based imaging set-up. The beam is modulated by a 2D grid before reaching the sample. The beam then propagates to the detector carrying information about the grid and the sample. Several images can be acquired with different positions of the grid.

The resulting sample image acquired multiplies the effect of the sample with the effect of the grid. Therefore, when going through Fourier transformation, the effects of the sample are convoluted with a 2D Dirac comb (Fourier transform of the mesh). Additionally, by isolating different harmonics of the Fourier transforms of the sample and reference images, it is possible to retrieve attenuation, scattering and differential phase contrast images of the samples. Figure 9 presents an example of a simulated sample image of a phantom with a grid and the associated Fourier transform. On the Fourier transform image, we can clearly see the different harmonics due to the mesh. Those harmonics are isolated by band pass filters, such as the blue and red circles drawn on the image, with radii ($r$) smaller than the separation width (SW) of the harmonics. After filtering the different ($m,n$) harmonics, their real space images $I_{s;m,n}(x,y)$ can be retrieved. As the mesh is not composed of perfect delta Dirac functions, those images can be corrected by the same harmonic image of the reference image $I_{r;m,n}(x,y)$ to obtain: $I_{m,n}(x,y) = I_{s;m,n}(x,y)/I_{r;m,n}(x,y)$. The information contained in that harmonic image can be written as:

$$I_{m,n}(x,y) = I_0(x,y)S_{m,n}(x,y)\exp(i\psi_{m,n}(x,y)) \tag{14}$$

where $I_0(x,y)$ is the intensity transmitted through the object, $S_{m,n}(x,y)$ is the real-valued scattering amplitude and $\psi_{m,n}(x,y) = Mz_2\mathbf{g}_{m,n}.\boldsymbol{\alpha}(x,y)$ is proportional to the phase derivative in the $\mathbf{g}_{m,n}$ direction. Here, $M$ is the magnification, $z_2$ is the propagation distance between the sample and detector and $\boldsymbol{\alpha}(x,y)$ is the refraction angle. From Equation (14), the sample transmission is obtained by taking the $0^{th}$ harmonic $I_0(x,y) = I_{0,0}(x,y)$. When taking the (0,1) and (1,0) harmonics, we get:

$$I_{0,1}(x,y)/I_0 = S_{0,1}(x,y)\exp(i\psi_{0,1}(x,y)) \tag{15}$$

Then, $S_{0,1}(x,y)$ is the norm of this complex ratio, and its angle gives $\psi_{0,1}(x,y)$ = $Mz_2\mathbf{g}_{0,1}.\boldsymbol{\alpha}(x,y) = Mz_2\alpha_x(x,y)$, from which the refraction along the $x$ axis $\alpha_x(x,y)$ can be easily extracted. The same treatment on the (1, 0) harmonic gives the refraction angle along the $y$ axis $\alpha_y(x,y)$. A more complex, even though quite similar harmonic-based method, also allows us to retrieve the dark-field signal.

This method was developed for conventional sources from the very beginning, first with simple 1D grids that gave only one phase derivative at a time [24,52], but it was rapidly adapted to 2D phase derivative by using meshes instead of simple grids [51]. It was also very rapidly used for in vivo imaging [53]. This methods appeared to be extremely promising for in vivo imaging and clinical transfer; however, it appears that very few works were carried out during the following years, and very little improvements to the technique were made. In 2019, it was proven that the technique gives quantitative phase images with a conventional source [54,55], and some works have been done to increase the retrieved images' resolution [56], which was then limited to the distance between two wires of the mesh.

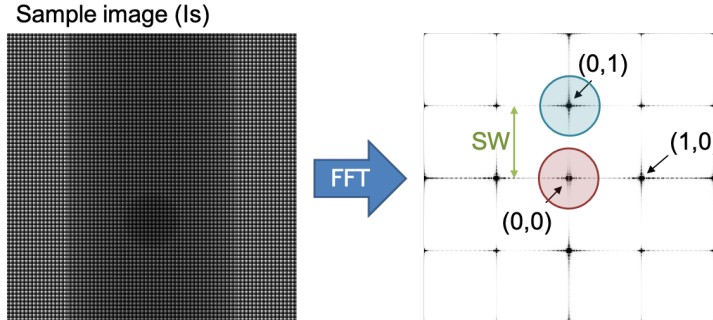

**Figure 9.** Mesh-Based imaging phase retrieval principle. Simulated sample image with a phantom and a grid. The Fourier transform of this image contains the Fourier transform of the sample convoluted to a Dirac comb. The harmonics contain various pieces of information about the sample.

### 3.6. Modulation-Based Imaging (Also Known as Speckle)

Random phase modulations techniques were developed with X-rays in 2012 [25,26] at synchrotrons, using the "speckle phenomenon" to create a random intensity pattern for which modifications upon introduction of the sample would give its phase information. In optics, speckles are random granular patterns that are produced by a coherent beam when deflected by an element with a rough surface (such as sandpaper at the synchrotron). With conventional sources with low or no coherence, higher Z materials have been used.

The experimental set-up of modulation-based imaging (MoBI) is shown in Figure 10.

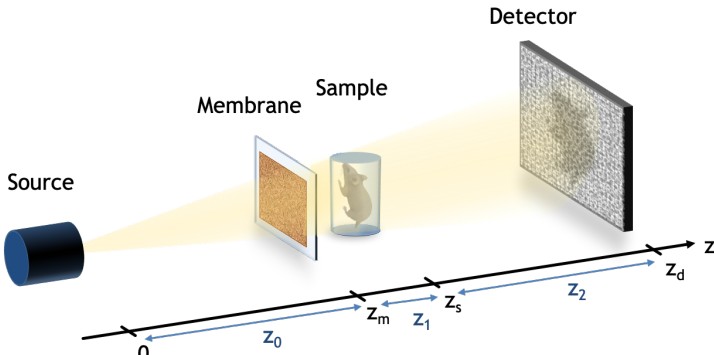

**Figure 10.** Modulation-based imaging set-up. The beam is modulated by a randomly structured membrane before reaching the sample. Then it propagates to the detector distorted by the membrane and the sample. Several images can be acquired with various positions of the membrane.

The experimental set-up only requires the sample, an X-ray imaging detector and a randomly structured membrane with structures a few pixels in size. The principle of MoBI is to follow the deflection of the beam's rays due to the object. To do so, a first image reference, $I_r$, is acquired with a random mask placed between the source and the detector, generating a random intensity modulations pattern, as shown on Figure 11. Then, a sample image, $I_s$, is acquired when an object is added in the path of the X-ray beam. MoBI is based on the tracking of the local distortions of the random pattern observed when comparing $I_s$ to $I_r$. This comparison allows one to retrieve the attenuation, refraction maps and dark-field signal. As mentioned before, the refraction is proportional to the phase derivative. The phase image ($\phi$) can then be retrieved by integrating the refraction maps. In order to improve the quality of the result, it is possible to take several pairs of acquisitions (with and without the sample), moving the membrane between each pair of acquisitions.

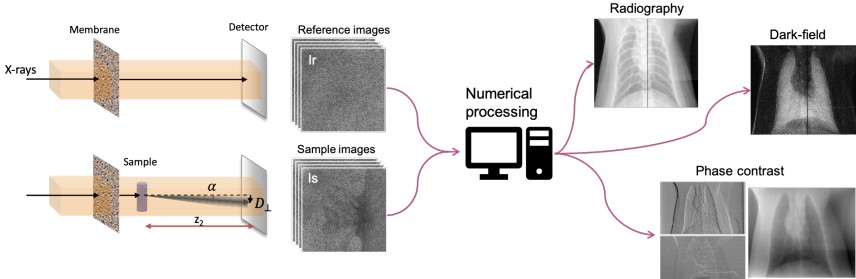

**Figure 11.** MoBI principle outline. From several acquisitions of the membrane alone and membrane plus sample in the path of the beam, numerical processing allows one to retrieve attenuation dark-field and phase contrast images.

This set-up, besides its simplicity of implementation, has the following main advantages:

- No field of view limitation other than the detector (the random mask can be easy to manufacture).
- No resolution limitation other than the optical system.
- Being radiation-dose-efficient because no absorbing element is used between the sample and the detector, meaning that all photons passing through the sample eventually contribute to the image formation.

For conventional systems, because the speckle phenomenon is difficult to produce, a new kind of membrane has to be invented using high-Z atomic elements (3D printed or powders deposition) to produce random intensity modulations [57]. Contrary to coded aperture systems [58] or EI that use binary masks (either letting the beam through unmodulated, or blocking it completely), the MoBI masks introduce an intensity modulation encoding each pixel of the image, meaning that sub-pixel precision is easier to obtain. Despite being easier to manufacture than coded aperture masks, and being easier to use (no need of precise alignment), due to the randomly introduced modulations being isotropic, it does not suffer from any frequency problems related to preferential direction detection as in mesh-based imaging and is sensitive to any directional structure.

To summarize, the experimental complexity of PCI and dark-field is translated in MoBI to the numerical processing side. This practical simplicity holds promise for the feasibility of a rotation gantry. Then, through an image analysis method, one can track the modulations displacements caused by the refraction of the sample directly in two directions. One conventional source phase can be retrieved using a system of linear equations [57]. Finally, modeled as multiple refraction [59], the dark-field signal can be extracted intrinsically using Fokker–Planck equations [60,61], or by tracking explicitly the modulation pattern distortions [62,63].

### 3.7. Recent Techniques

Very recently, new methods in between MoBI and EI have emerged based on Hartmann wavefront sensors [64] or beam tracking [65]. These techniques use a highly absorbing plate with punched holes to split the beam into very thin beamlets and then track their displacements and distortions on the detector plane. The main difference with MoBI is that the sample is not entirely illuminated at once due to the binary "modulation" of the beam. These methods, though very recent and therefore still rare in the literature, appear to have various advantages and were already proven to work with a laboratory liquid-metal jet source [66].

### 4. Conclusions

The phase-sensitive techniques described above differ in their technical complexity, the use of X-ray optical elements, the readout process and the method of extracting the signals to detect and to measure the phase contrast and dark field. Thus, all these modalities face various limitations for the transfer to conventional systems. PBI and ABI both require



highly coherent sources (mainly spatial coherence for PBI and temporal coherence for ABI), and very few works mention their use for extracting the dark-field signal in the literature. In addition, ABI requires a complex set-up with a high flux, and is therefore difficult to implement with conventional X-ray tubes. PBI has a very simplistic set-up, but the source coherence requirement makes its transfer to low-cost X-ray source devices hard.

GI, EI, MBI and MoBI have proven to be transferable to conventional sources; however, GI and EI have quite complex set-ups and are only sensitive to one direction of refraction that causes bad performance in terms of noise power spectrum for tomographic reconstruction [46]. Moreover, the presence of numerous optical elements, in GI and EI setups, yields as long as 7 s acquisition time for chest radiography [43]. On the contrary, MBI and MoBI have a simple set-up sensitive to two aspects. MBI and MoBI, despite their limited sensitivity, have the advantage of a high simplicity of implementation that might facilitate a transfer of phase contrast and dark field on clinical devices with gantries that rotate around a sample or a patient.

Finally, to enhance the quality of phase and dark-field retrieval maps, artificial intelligence algorithms are starting to appear [67,68] for different set-ups that enhance the quality of the images and/or reduce virtually the radiation dose deposited to the sample. This could tackle the last challenges of phase contrast and dark-field transfer in conventional systems for clinics.

**Author Contributions:** All authors conceived and designed the analysis, collected the litterature data and wrote the paper. All authors have read and agreed to the published version of the manuscript.

**Funding:** This work was performed within the framework of LABEX PRIMES (ANR-11-LABX-0063) of Université de Lyon, within the "Investissements d'Avenir" program (ANR-11-IDEX-0007) operated by the French National Research Agency (ANR).

**Institutional Review Board Statement:** Not applicable.

**Informed Consent Statement:** Not applicable.

**Data Availability Statement:** Not applicable.

**Conflicts of Interest:** The authors declare no conflict of interest.

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
