# Peer review of "X-ray Phase Contrast Imaging from Synchrotron to Conventional Sources: A Review of the Existing Techniques for Biological Applications"

_applsci, doi:10.3390/app12199539_

Round 1
Reviewer 1 Report
1. The abstract needs modification. However, the significant conclusions of the paper must be briefly mentioned at the last paragraph of abstract. Need modification.
. More specification of governing equations is needed with reference.3. A working flowchart is required to understand the whole process.
4. Modify the ‘Conclusion’ section. It should be specific. Restate the hypothesis briefly and summarize the key findings throughout with further applications.
Author Response
We thank the referee for the useful comments. Here is a point-by-point answer.
First of all the revision of our manuscript was checked and corrected by an English native speaker.
. The abstract needs modification. However, the significant conclusions of the paper must be briefly mentioned at the last paragraph of abstract. Need modification.
The abstract was modified and a reminder of the conclusion is included now
. More specification of governing equations is needed with reference.
More references have been added to the manuscript. Our equation can be found in only one book, which is the reference in x-ray phase contrast imaging, and this book is cited several times now.
A working flowchart is required to understand the whole process.
We do no understand this remark and don't know which is the process the referee is referring to.
Modify the ‘Conclusion’ section. It should be specific. Restate the hypothesis briefly and summarize the key findings throughout with further applications.
The conclusion was modified with more key finding added.
Reviewer 2 Report
1. The topic of this manuscript focuses on the Phase Contrast Imaging (PCI) techniques those adapted on conventional X-ray sources as the title says, however, it spends a consider part on the introduction of other X-ray imaging methodologies. This part can be further simplified.
2. In the PCI reviewing part, more introduction and emphasis is placed on the theoretical basis, compared to the introduction of instrumentation for each technique. It is recommended to add more introduction briefly on the relevant equipment setup and the X-ray sources requirements for each technique.
3. The abbreviations of professional terms should be unified. For example, the term "Dark-field Imaging" is noted as DI in line 19, while as DFI in line 131, and as DF in line 163.
Author Response
We thank the referee for the remarks that improved the manuscript.
Here is point-by-point answer to your remarks:
The topic of this manuscript focuses on the Phase Contrast Imaging (PCI) techniques those adapted on conventional X-ray sources as the title says, however, it spends a consider part on the introduction of other X-ray imaging methodologies. This part can be further simplified
We believed that the governing equations are essential for the understanding of the rest of the manuscript; therefore we did not modify the concerned section.
In the PCI reviewing part, more introduction and emphasis is placed on the theoretical basis, compared to the introduction of instrumentation for each technique. It is recommended to add more introduction briefly on the relevant equipment setup and the X-ray sources requirements for each technique.
Details of the instrumentation of each technique has been added.
The abbreviations of professional terms should be unified. For example, the term "Dark-field Imaging" is noted as DI in line 19, while as DFI in line 131, and as DF in line 163.
This is now corrected and only one acronym is used
Reviewer 3 Report
The manuscript reviews the techniques of X-ray phase contrast imaging that use Synchrotron light source or tube set-ups. In the referee’s opinion it is not clear which is the real intention of the report and why it should be relevant to read such a review to have an overview of the present technology. The paper is deficient to clarify which applications can be targeted with such kinds of technologies and what is motivating the scientific community to invest in not trivial research and development. It looks like the paper is unbalanced with a very long and detailed introduction about the physical mechanism and theoretical background of phase contrast and a too short description of the actual implementations of the various techniques. In this respect, it is not clear if the main application of PC is medical for in vivo for which the dose required to realize useful images is relevant, or ex vivo, for which the dose is less critical but the long time scans can be problematic for the sample stabilization.
On the other side, there are several material science applications that can profit from phase contrast and dark fields, which motivated several research investments, both public and private, and they are not mentioned at all. Another aspect that is extremely relevant for the realization of the various techniques is the instrumentations, which is a very active field of research and technology that has not been described at all (gratings, X-ray sources, detectors, geometries). Therefore, the review looks very poor and incomplete and it would require a severe revision in order to be acceptable. The referee strongly recommend to consider the following list of references, which support the above referee opinion:
Applications
www.https://doi.org/10.1016/j.jsb.2014.08.003
https://www.nature.com/articles/s41598-021-97915-y
www.http://www.opticsexpress.org/abstract.cfm?URI=oe-29-2-2049
www.https://doi.org/10.1016/j.compositesb.2022.109634
www.https://doi.org/10.1038/s41598-021-93054-6
www.https://doi.org/10.1007/s00330-019-06362-x
Techniques
www.http://opg.optica.org/optica/abstract.cfm?URI=optica-8-12-1538
www.http://dx.doi.org/10.1038/srep35259
https://aip.scitation.org/doi/10.1063/5.0087940
www.https://doi.org/10.1038/srep05198
www.https://doi.org/10.1038/s41598-022-05965-7
www.https://doi.org/10.1002/ima.22520
X-ray optics fabrication
www.https://doi.org/10.1007/s00542-008-0584-5
www.http://scitation.aip.org/content/aip/proceeding/aipcp/10.1063/1.4742267
www.https://doi.org/10.1117/1.JMM.20.4.043801
www.http://ol.osa.org/abstract.cfm?URI=ol-46-15-3693
www.https://doi.org/10.1016/j.apsusc.2022.152938
www.http://dx.doi.org/10.1149/1945-7111/abba63
www.http://avs.scitation.org/doi/abs/10.1116/1.4991807
www.https://www.mdpi.com/2072-666X/11/6/589
www.http://dx.doi.org/10.1039/C9NH00709A
www.https://onlinelibrary.wiley.com/doi/abs/10.1002/adem.202000258
www.https://www.mdpi.com/2072-666X/11/9/864
Minor details
1. Please keep the acronym consistent in the whole text: phase contrast is sometimes indicated as PC, PCI; dark field is DF, DFI etc.
2. Line 285 “teem” should be “team”
3. The difference between an absorbing grating G2 in GI, G0 in Edge and a 2D grid is not clear, the authors should describe these optics in details and maybe indicating some typical feature size otherwise it is almost impossible for a reader that is not in the field to understand the difference among the various set-ups.
Author Response
We thank the reviewer for her/his comments that improved our manuscript and for list of references we were aware about. Before going to a point by point answer to the referee's remarks here is a general remark:
Our intention with this review was to give an overview of the different techniques for biological applications where the dose and acquisition time are critical parameters. To make us clearer the title has been changed. Moreover the goal of this paper is not to cite huge range of application or technical development of PCI in domains not related to biology or medicine that would to our opinion lead to some loss of clarity, indeed we think that all major renown expert in this domain have been cited, the objective being to make a rapid overview intelligible to non expert in the field of Phase Contrast and to guide them into the different available developed techniques of PCI. That is also why the theoretical foundation of PCI is long according to the reviewer (3/17 pages) to let people understand the physical mechanisms. The focus was not note on the applications because one can find existing excellent reviews on the applications of PCI and dark field (Bravin et al PMB 2012, Birnbacher, L., Braig, EM., Pfeiffer, D. et al. Eur J Nucl Med Mol Imaging (2021), for instance). To our opinion, what was missing from the literature is a review of the techniques with the mention of Mesh Based Imaging and Speckle (Modulation) based Imaging.
Here is a point by point answer to the reviewer's remarks:
The manuscript reviews the techniques of X-ray phase contrast imaging that use Synchrotron light source or X-ray tube set-ups. In the referee’s opinion it is not clear which is the real intention of the report and why it should be relevant to read such a review to have an overview of the present technology.
We give detail explanation in the general remark above
The paper is deficient to clarify which applications can be targeted with such kinds of technologies and what is motivating the scientific community to invest in not trivial research and development:
The first sentence of the introduction clearly suggests the interest of PCI and why the biomedical scientific community is investing in such technology
looks like the paper is unbalanced with a very long and detailed introduction about the physical mechanism and theoretical background of phase contrast and a too short description of the actual implementations of the various techniques. In this respect, it is not clear if the main application of PC is medical for in vivo for which the dose required to realize useful images is relevant, or ex vivo, for which the dose is less critical but the long time scans can be problematic for the sample stabilization.
We give detail explanation in the general remark above
“On the other side, there are several material science applications that can profit from phase contrast and dark fields, which motivated several research investments, both public and private, and they are not mentioned at all. Another aspect that is extremely relevant for the realization of the various techniques is the instrumentations, which is a very active field of research and technology that has not been described at all (gratings, X-ray sources, detectors, geometries). Therefore, the review looks very poor and incomplete and it would require a severe revision in order to be acceptable. “
Already answered in the general remark.
The referee strongly recommend to consider the following list of references, which support the above referee opinion
To our opinion, the proposed list is strongly biased with 19 articles out of 23 coming from the same team, that appears to us quite strange and not really standard, as said we take care to provide non-expert readers fair references of all renown expert in the field to provide the best overview for the readers. Moreover some articles are completely out of the scope of the present review (artificial intelligence based segmentation; numerical simulations; techniques that were adapted to conventional systems). Nevertheless we added a sentence on the fabrication of gratings with fair citations representing the community to our opinion.
The added sentence is the following:
“The fabrication of Gratings Interferometer is a very challenging task and is a very active field of research \cite{pinzek2021fabrication,romano2020high,noda2008fabrication} which is not the purpose of this review. ”
Minor details
Everything has been corrected accordingly to the reviewer's remarks.
Round 2
Reviewer 3 Report
The referee thanks the authors for discussing the critical points, it is authors' decision which reference to cite and which criteria to use in the selection. The referee suggested a broader range of references, the authors choice is also biased citing more papers from a specific group. The referee does not want to enter in this debate and let the editor deciding. In such cases, it would be a good practice to look in details and see which reference has been submitted first.